

# Image-based effective feature generation for protein structural class and ligand binding prediction

Nafees Sadique*, Al Amin Neaz Ahmed*, Md Tajul Islam,
Md. Nawshad Pervage and Swakkhar Shatabda

Department of Computer Science and Engineering, United International University, Dhaka,
Bangladesh
* These authors contributed equally to this work.

Corresponding author
Swakkhar Shatabda,
swakkhar@cse.uiu.ac.bd

## ABSTRACT

Proteins are the building blocks of all cells in both human and all living creatures of the world. Most of the work in the living organism is performed by proteins. Proteins are polymers of amino acid monomers which are biomolecules or macromolecules. The tertiary structure of protein represents the three-dimensional shape of a protein. The functions, classification and binding sites are governed by the protein's tertiary structure. If two protein structures are alike, then the two proteins can be of the same kind implying similar structural class and ligand binding properties. In this paper, we have used the protein tertiary structure to generate effective features for applications in structural similarity to detect structural class and ligand binding. Firstly, we have analyzed the effectiveness of a group of image-based features to predict the structural class of a protein. These features are derived from the image generated by the distance matrix of the tertiary structure of a given protein. They include local binary pattern (LBP) histogram, Gabor filtered LBP histogram, separate row multiplication matrix with uniform LBP histogram, neighbor block subtraction matrix with uniform LBP histogram and atom bond. Separate row multiplication matrix and neighbor block subtraction matrix filters, as well as atom bond, are our novels. The experiments were done on a standard benchmark dataset. We have demonstrated the effectiveness of these features over a large variety of supervised machine learning algorithms. Experiments suggest support vector machines is the best performing classifier on the selected dataset using the set of features. We believe the excellent performance of Hybrid LBP in terms of accuracy would motivate the researchers and practitioners to use it to identify protein structural class. To facilitate that, a classification model using Hybrid LBP is readily available for use at http://brl.uiu.ac.bd/PL/. Protein-ligand binding is accountable for managing the tasks of biological receptors that help to cure diseases and many more. Therefore, binding prediction between protein and ligand is important for understanding a protein's activity or to accelerate docking computations in virtual screening-based drug design. Protein-ligand binding prediction requires three-dimensional tertiary structure of the target protein to be searched for ligand binding. In this paper, we have proposed a supervised learning algorithm for predicting protein-ligand binding, which is a similarity-based clustering approach using the same set of features. Our algorithm works better than the most popular and widely used machine learning algorithms.

## INTRODUCTION

Protein tertiary structure comparison is very important in many applications of modern structural biology, drug design, drug discovery, in studies of protein-ligand binding, protein-protein interactions, and other fields. This is especially significant because the structure of a protein is more protected than the protein sequence (*Chothia & Lesk, 1986*). Many works have been done to find protein binding (*Brady & Stouten, 2000*). Comparison of protein structure has been done in many works of literature by alignment of distance matrices (*Holm & Sander, 1993*), using iterated double dynamic programing (*Taylor, 1999*), using elastic shape analysis (*Srivastava et al., 2016*) and many other techniques. The most common way of comparing protein tertiary structure is to treat the protein as a three-dimensional object and superimpose one on another. Different distances are used to calculate the differences between the proteins.

The distance matrix of α carbon can be seen extensively used in *Holm & Sander (1997)* and *Singh & Brutlag (1997)* as a feature which represents the tertiary structure of a protein chain. This feature is used as a feature vector which represents the structure of a protein to measure either similarity or dissimilarity to measure and compare the feature vectors with one another in pattern recognition literature. A mapped two-dimensional feature matrix is created from the 3D coordinate data of protein. The intra-molecular distance is used to make the α carbon distance matrix which mirrors the tertiary structure of a protein and the conserved elements of the secondary structure in it. With an input matrix size of N × N, the distance matrix based exact algorithms run in $O(N)$ time (*Karim et al., 2015*).

An image is basically a matrix of N × N dimension with corresponding data in each cell. Thus the distance matrix can be used as an image. Basically, three types of features can be generated from an image: pixel-based, filter-based and computationally generated features. Pixel-based features, for example, histograms are simplistic and dependent on the capability of classification algorithms. Filter-based methodologies transform the original image to use feature extraction methods. Refined algorithms are used to segment and other various algorithms are used to detect different features. Using ideas from computer vision and utilizing it in protein structure retrieval is not uncommon in the field. ProteinDBS server (*Shyu et al., 2004*) implement a similar approach in *Chi, Scott & Shyu (2005)*. Texture features from the original size images and diagonally partitioned images were extracted by *Chi, Scott & Shyu (2005)*. CoMOGrad and PHOG (*Karim et al., 2015*) also used images to extract their two novel feature whereas we are extracting histograms of local binary pattern (LBP) images from the original image.

The human body uses protein for repairing tissues, making enzymes, hormones, and other biological chemicals. It is an essential building block of bones, muscles, cartilage, skin, and blood. On the other hand, a ligand is a material that has the potentiality to bind

to and forms a composite with a biomolecule in order to carry out a biological function. In protein-ligand binding, the ligand is usually a molecule which produces a signal by binding to a locus on a target protein. The binding typically results in a change of conformational isomerism (conformation) of the target protein. The evolution of the protein's responsibility depends on the development of specific sites which are designed to bind ligand molecules. Ligand binding ability is important for the management of biological functions. Ligand binding interactions change the protein state and function. Protein-ligand binding prediction is very important in many applications of modern structural biology, drug design, drug discovery, and other fields.

Many experimental techniques can be used to investigate various aspects of protein-ligand binding. X-ray crystallography, nuclear magnetic resonance, Laue X-ray diffraction, small-angle X-ray scattering and cryo-electron microscopy provide atomic-resolution or near-atomic-resolution structures of the unbound proteins and the protein-ligand complexes, which can be used to study the changes in structure and/or dynamics between the free and bound forms as well as relevant binding events. Although experimental techniques can investigate thermodynamic profiles for a ligand-protein complex, the experimental procedures for determination of binding affinity are laborious, time-consuming, and expensive. Modern rational drug design usually involves the HTS of a large compound library comprising hundreds or thousands of compounds to find the lead molecules, but this is still not realistic to use experimental methods alone. Different methods like isothermal titration calorimetry (*Chaires, 2008*), surface plasmon resonance (*Patching, 2014*), fluorescence polarization (*Rossi & Taylor, 2011*), protein-ligand docking (*Sousa et al., 2013*), free energy calculations (*Steinbrecher & Labahn, 2010*), etc., are being used to predict ligand-binding prediction.

In this paper, we propose the combination of LBP histogram, Gabor Filtered LBP Histogram, Separate Row Multiplication Matrix with Uniform LBP Histogram, neighbor Block Subtraction Matrix with Uniform LBP Histogram and Atom Bond features to be used for protein similarity measurement. We extract the distance matrix of α carbon of a protein from PDB file and use the distance matrix as an image to extract our first four features, and the atom bond is extracted from the PDB files. We have used a large variety of classification algorithms to test the extracted features. We also show the results and comparative study of different implementation methodology of CoMOGrad and PHOG. The method we have proposed can produce a better result on some classification algorithm over the previous methods on the same benchmark. In addition, we have proposed a supervised learning algorithm for predicting Protein-Ligand Binding which is a similarity-based clustering approach using the same set of features. Our algorithm works better than the most popular and widely used machine learning algorithms. Our proposed method uses the features proposed in this paper.

## MATERIALS AND METHODS

Our methodology is divided into two parts. Firstly, we have generated image-based features using protein tertiary structures and performed feature analysis based on the prediction power on the structural class prediction problem. In this section, we present the

materials and methods for both of the problems. The dataset, features, necessary algorithms and performance measurements are described accordingly for each of the problems.

## Structural class prediction

In this section, we present the methodology of structural class prediction. Atom bond features are generated from the protein tertiary structures given as PDB files. Images are created from the distance matrix calculated using α carbon atom coordinates of the amino acids of the protein structures in the given dataset. From each image of protein, we have derived five features. Stratified Remove Folds was used to test the capability and efficiency of the dataset. There are in total seven different classes of protein structures. Synthetic minority over-sampling technique (SMOTE) is used to handle class imbalance problem. The block diagram of the methodology is given in Fig. 1.

### Structural class prediction dataset

We have used 40 percent ID filtered subset of PDB-style files for SCOPe domains version 2.03 (*Fox, Brenner & Chandonia, 2013*) as our dataset. It contains a total of 12,119 PDB files. Each PDB files contains SCOP(e) concise classification string (sccs) which respectively describes class, fold, superfamily, and family. In this paper, we are going to experiment only with the class of the protein. In the dataset, there are total seven protein structural classes. For benchmark analysis with CoMOGrad and Phog, the common PDB files were used as dataset. The common PDB files are total of 11,052. The details of the protein structural classes are given in Table 1. This dataset is widely used as a benchmark in the literature for protein structural similarity prediction (*Karim et al., 2015*).

### Image generation

We have generated images of protein structures according to the methodology described in CoMOGrad and PHOG (*Karim et al., 2015*). Only α carbons of the amino acids in the protein structure are considered for image generation. From the three dimensional coordinates of the α carbon atoms, a distance matrix is generated by taking the Euclidean distance among all pairs. This distance matrix converts a 3D structure of a protein to a 2D matrix. Euclidean distance is applied as the distance measure because it exceeds the popularly applied costly alignment distance measure of α carbon distance matrices. Thus only half of the matrix contains redundant information due to symmetry. The matrix can be further regarded as an image.

### Scaling of images

The dimension of protein images is based on the total number of α carbon they have. So, every individual protein images are different from the other in dimension. Therefore, the images were scaled to the same dimension. CoMOGrad and PHOG have used Bi-cubic interpolation and wavelet transform to scale all the protein images into $128 \times 128$ dimension (*Karim et al., 2015*). During the Bi-cubic interpolation step, most of the images were in $128 \times 128$ dimension so in the wavelet transform step they scaled all the images to

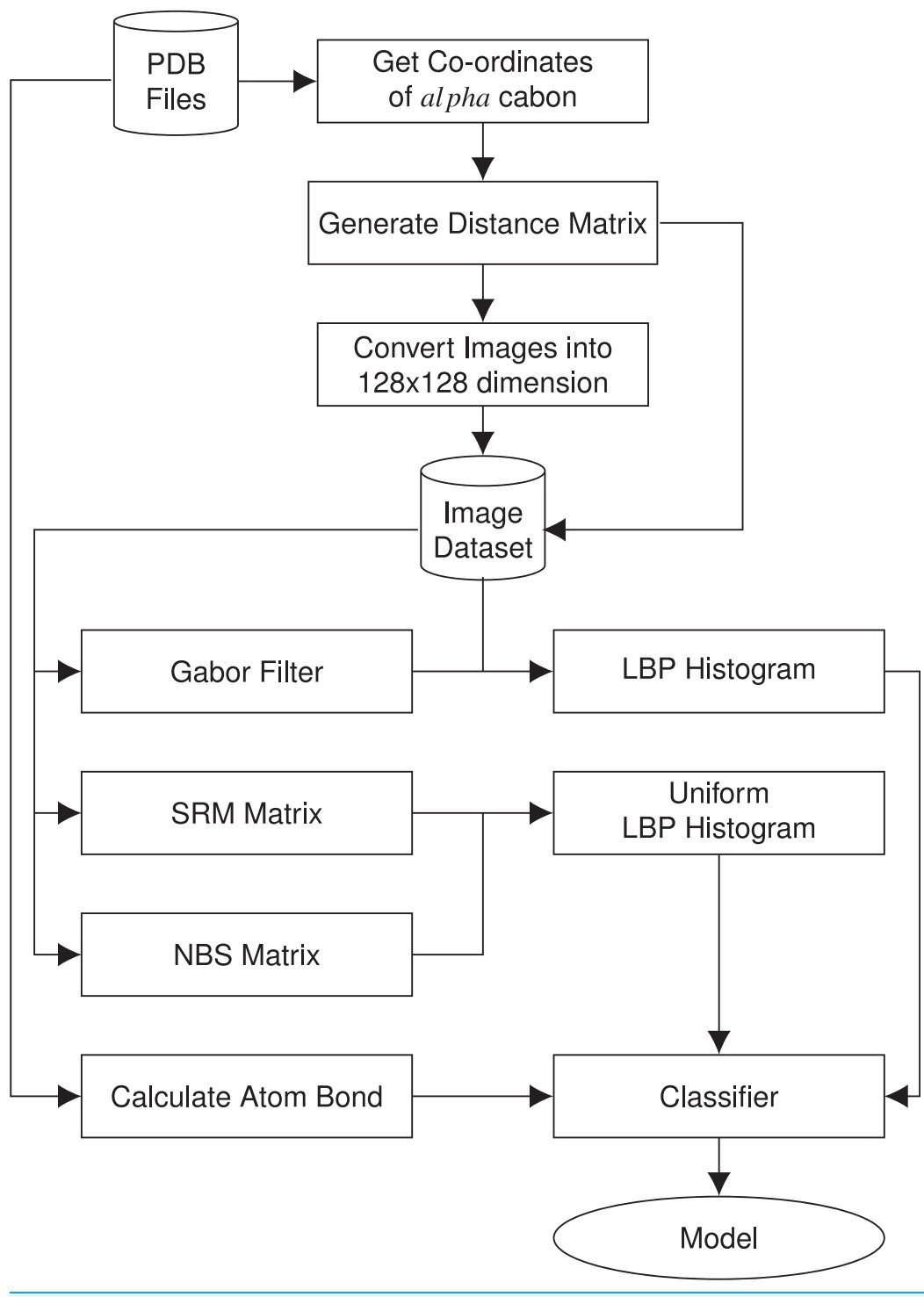

**Figure 1** **Block diagram of the methodology used in structural class prediction.**

that dimension. Thus, we have directly scaled the images to $128 \times 128$ dimension. We have used both real and scaled images to examine the differences in their predictive power. Sample rescaled images of protein structures are given in Fig. 2.

**Table 1 Protein classes and its corresponding instances.**

| Class name | Total instances |
| --- | --- |
| Small proteins | 640 |
| All α proteins | 2,195 |
| α and β proteins (a/b) | 3,305 |
| α and β proteins (a + b) | 3,006 |
| Membrane and cell surface proteins and peptides | 204 |
| All β proteins | 1,485 |
| Multi-domain proteins (α and β) | 219 |

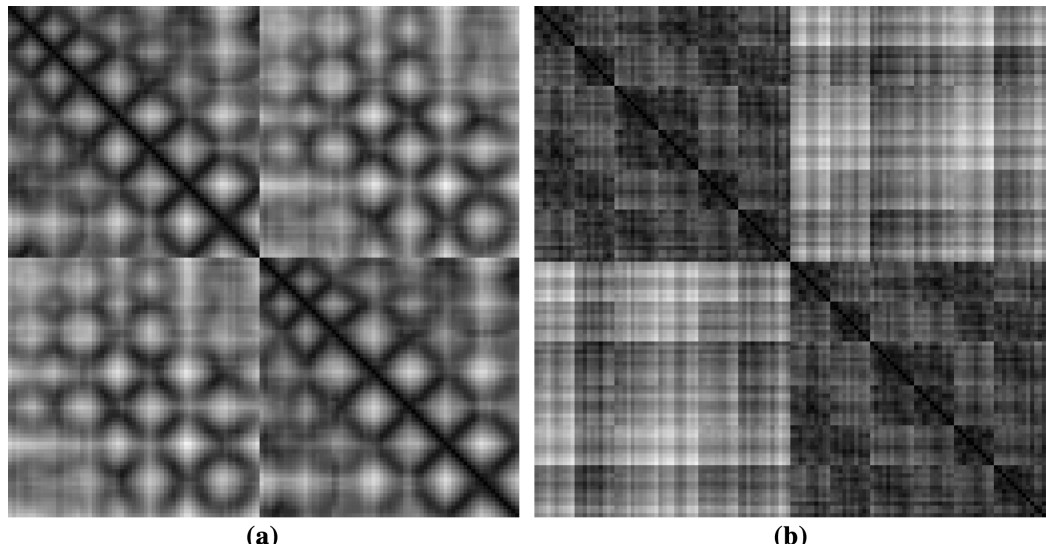

(a)  (b)

**Figure 2 Sample images of protein structures after rescaling (A) showing diagonal and (B) symmetric textures.**

### Feature extraction

We have generated five different feature groups. Our first four feature groups are different types of histograms and the fifth feature group is about the prognosis of the atoms. The histograms were taken from both scaled and unscaled images.

### Local binary pattern histogram

Local binary pattern histogram was first proposed by *Ojala, Pietikainen & Harwood (1994)* and popularized by the work of *Ojala, Pietikainen & Maenpaa (2002)*. LBP computes the local representation of the texture of an image as a texture descriptor. Comparing each pixel with its neighboring pixels the local representation is created. The image is transformed into a grayscale image. In a 3 × 3 neighborhood, the center pixel value is calculated by comparing with its eight neighboring pixels. Each comparison gives a result of either 0 if the center pixel value is greater than the comparing neighbor pixel or 1 for the latter. A clockwise direction starting from the top-left one provides a binary number. The binary number is converted to a decimal number and the value is placed in the center pixel. LBP codes or LBPs are the obtained binary numbers. An example of a

| 6 | 5 | 2 | | | 1 | 1 | 0 |
|---|---|---|---|---|---|---|---|
| 9 | 4 | 2 | Threshold → | | 1 | | 0 |
| 1 | 7 | 8 | | | 0 | 1 | 1 |

Binary : 11001101
(ClockWise)
Decimal : 205

**Figure 3 An example of basic LBP.** 

basic LBP is given in Fig. 3. After calculating the value for each pixel of the image, a histogram is calculated. A $3 \times 3$ neighborhood has $2^8 = 256$ possible patterns, thus the values range from 0 to maximum 255 in each pixel of the image. The total number of bins of the histogram is thus 256. We would get 256 attributes from each image. We have used zero-padding technique to generate LBP.

### Gabor filtered local binary pattern histogram

Gabor Filter is titled after Dennis Gabor. It is used for texture segmentation (*Jain & Farrokhnia, 1991*), optical character recognition (*Jain & Bhattacharjee, 1992*), edge detection (*Mehrotra, Namuduri & Ranganathan, 1992*), and so on. It is a linear filter that examines if there is any particular frequency content in the image in specific areas in a localized region throughout the point. In spatial and frequency domain, the Gabor filter has been determined to own optimal localization properties (*Jain, Ratha & Lakshmanan, 1997*). Gabor filter lets a particular band of frequencies while discarding others and that is why it is designated as a bandpass filter. It resembles the features of simple visual cortical cells (*Dunn & Higgins, 1995*). It provides the largest response where texture changes at edges and at points. The multiplication of a sinusoid and a Gaussian is called the Gabor filter. The filter has a real and an imaginary segment rendering orthogonal directions. The two segments can be set into a complex number or used individually. Various shapes, sizes and smoothness levels in an image can be detected by Gabor filter.

$$g(x, y; \lambda, \theta, \phi, \gamma) = \exp\left(-\frac{x'^2 + \gamma^2 y'^2}{2\sigma^2}\right) \exp\left(\imath\left(2\pi\frac{x'}{\lambda} + \phi\right)\right) \tag{1}$$

$$g(x, y; \lambda, \theta, \phi, \gamma) = \exp\left(-\frac{x'^2 + \gamma^2 y'^2}{2\sigma^2}\right) \cos\left(2\pi\frac{x'}{\lambda} + \phi\right) \tag{2}$$

$$g(x, y; \lambda, \theta, \phi, \gamma) = \exp\left(-\frac{x'^2 + \gamma^2 y'^2}{2\sigma^2}\right) \sin\left(2\pi\frac{x'}{\lambda} + \phi\right) \tag{3}$$

Equation 1 is the complex version of the Gabor function and Eqs. (2) and (3) are real and imaginary version respectively. The shape and size of the Gabor function is regulated by its five parameters. Here, $\lambda$ controls the wavelength of this sinusoid, $\theta$ is the angle of the normal to the sinusoid, $\phi$ is the phase shift of the sinusoid, $\gamma$ controls the aspect ratio, The spatial envelope or the standard deviation of the Gaussian is $\sigma$. For our experiments, we have used $\lambda = 10$, $\theta = 0$, $\phi = 0$, $\gamma = 0.02$ and $\sigma = 5$. After applying the Gabor filter, LBP techniques are applied to the image to get 256 attributes.

### Atomic bond features

First of all, we've identified unique atoms amidst all the protein PDB files. From each protein PDB file, we've counted occurrences of each atom. Then we've taken the percentage as features of each atom among all the atoms that each protein has. Then we've taken the first 100 sequential atoms and used their atomic mass as the feature. Then we've counted the bond that each pair of atoms has in a particular protein using atomic distance based on a threshold value. Finally, we've taken the percentage as the feature of the bond of each unique pair of atoms among all the bonds that the protein has.

### Separate row multiplication matrix with uniform LBP histogram

The image is split into $3 \times 3$ matrices. From each matrix, we get three rows with the dimension of $1 \times 3$. By multiplying each row with the same $3 \times 3$ matrix, we get three result matrix consisting of $1 \times 3$ dimension. Each cell is divided by 100. The results are then put in the $3 \times 3$ matrix in accordance with the row numbers. The color intensity of an image is between 0 to 255. So, if the value of any cell of the result matrix is greater than 255, then the value is replaced with 255. After applying this technique, the uniform LBP is applied. From Fig. 4, (a) presents a $3 \times 3$ section of matrix and the rows, (b) exhibits the result of multiplication, (c) shows the value after dividing by 100, (d) shows the replacement result of value greater than 255 and (e) shows a $3 \times 3$ matrix section after SRM-Matrix transformation.

Another variation of the LBP is called uniform pattern (*Ojala, Pietikainen & Maenpaa, 2002*). Some binary patterns occur more generally in texture images. If the binary pattern comprises at most two 0-1 or 1-0 transitions when the bit pattern is held circular then the pattern is called uniform. For instance, 01000000 has two transitions, 00000111 has two transitions which are uniform pattern on the other hand 01010100 has six transitions,11001001 has four transitions which are not uniform. A neighborhood with a dimension of $3 \times 3$ has $2^8 = 256$ possible patterns with 58 of them being uniform. For estimating the histogram, every uniform pattern gets a separate bin while a single bin is allotted for all non-uniform patterns. Therefore, from a uniform binary pattern, we get the histogram of the total bin size of 59.

### Neighbor block subtraction matrix with uniform LBP histogram

Blocks are of the same dimension, $3 \times 3$. Two blocks of matrices are considered neighbors for this method if the center cells are neighboring. Because of this, the value of the last two columns of the first block and the first two columns of the second block are the same. The two blocks of matrices are subtracted and the result is set in the place of the first block. If any of the cells have any negative number, then 0 is placed instead of the negative value. The replacement of value is made because the histogram bin begins from zero. Uniform LBP is then used to compute the histogram.

Summary of all the feature groups used in this paper is given in Table 2.

### Handling imbalance in data

From Table 1 it can be noted that the classes are imbalanced. To balance the classes, we have used SMOTE (*Chawla et al., 2002*). The percentage of SMOTE indicates that how

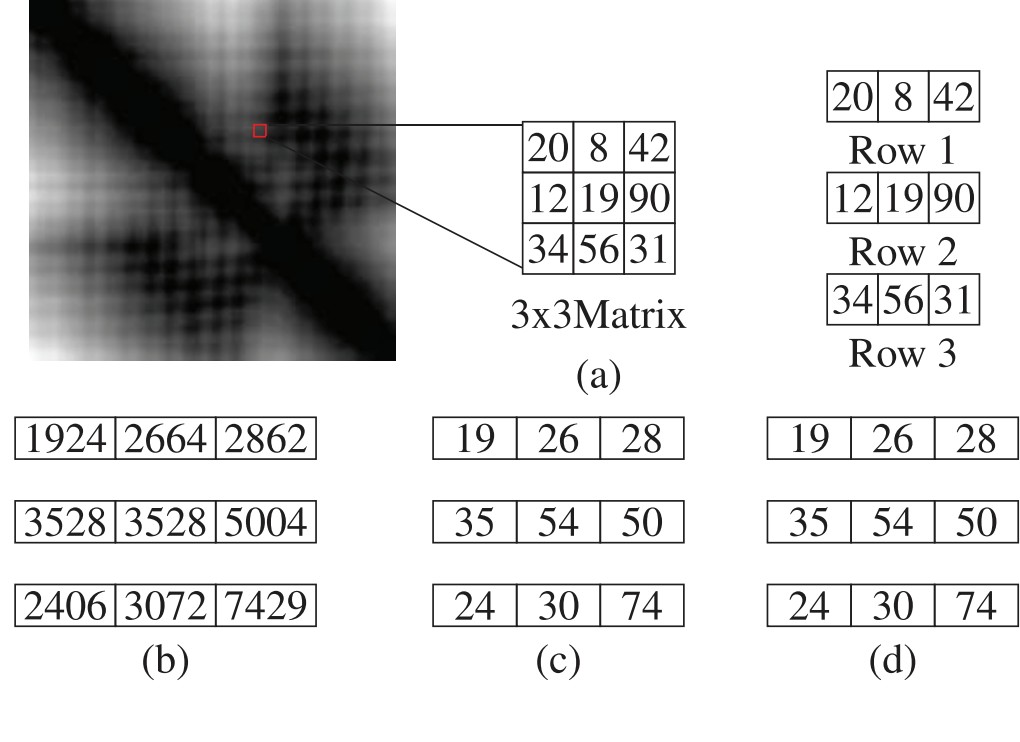

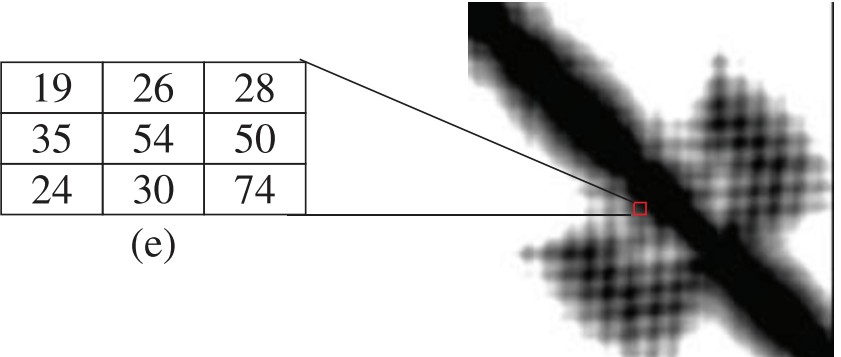

**Figure 4 An example of separate row multiplication matrix with uniform local binary pattern histogram (A) sample figure and matrix; (B) after multiplication; (C) and (D) separation of the matrices and (E) finally showing the filtered image.**

**Table 2 Feature groups.**

| Identifier | Feature group name | Number of features |
|---|---|---|
| A | LBP-Hist | 256 |
| B | GfLBP-Hist | 256 |
| C | Atom Bond | 116 |
| D | SRMMat-ULBP-Hist | 59 |
| E | NBSMat-ULBP-Hist | 59 |

many more instances would be generated. We have over-sampled our instances close to the highest number of instances among all the classes. If $x$ denotes the highest number of instances among all the classes and $y$ is denoted by a class which we will SMOTE, then the expression for the percentage calculation is $\frac{x-y}{y} * 100$. We have used five nearest neighbors to generate the over-sampled instances.

It is to be noted that SMOTE was used on only train dataset after separating the train and test dataset. So, there are no artificial instances in the test dataset.

### Classifiers used

We have used five classifiers for the analysis of features applied to solve structural class prediction problem: K-Nearest Neighbor (KNN), Naive Bayesian Classifier, support vector machines (SVM), Adaptive Boosting (AdaBoost) and Random Forest. A concise description of the classifiers is given in this section.

### K-nearest neighbor

K-nearest neighbor algorithm (KNN) (*Aha, Kibler & Albert, 1991*) is a similarity-based classification technique. It is a lazy classification technique. Distance metrics are used for each instance of the whole dataset for calculating the KNN. The labels of the nearest neighbors decide the label of the test instances. It works poorly for high dimensional data. Euclidean distance, Hamming distance, Manhattan distance, Minkowski distance, Tanimoto distance and Jaccard distance are used for similarity measures.

### Naive Bayesian classifier

Naive Bayesian classifier (*Maron, 1961*) is based on probabilistic inference of samples observed where the decision variable and the features form a very naive structure of Bayesian Network. Naive Bayesian classifiers work best for image recognition and text mining.

### Support vector machine

Support vector machine (*Cortes & Vapnik, 1995*) works by creating and separating hyperplane for a given dataset by sampling different classes which are separated by maximum width.

### Adaptive boosting

Adaptive Boosting classifier (*Freund & Schapire, 1997*) is a meta-classifier which aims to make a strong classifier using a set of weak classifiers. The classifiers whose performance is marginally better than random classifiers are called weak classifiers.

### Random forest

Random Forest (*Ho, 1995*) is an ensemble classifier. A decision tree is created in each iteration with features taken randomly. It samples selected features using bootstrap aggregating.

## Ligand-binding prediction

Protein-ligand binding prediction is a binary class classification problem. We have used image-based features for each protein and ligand dataset. Our methodology learns

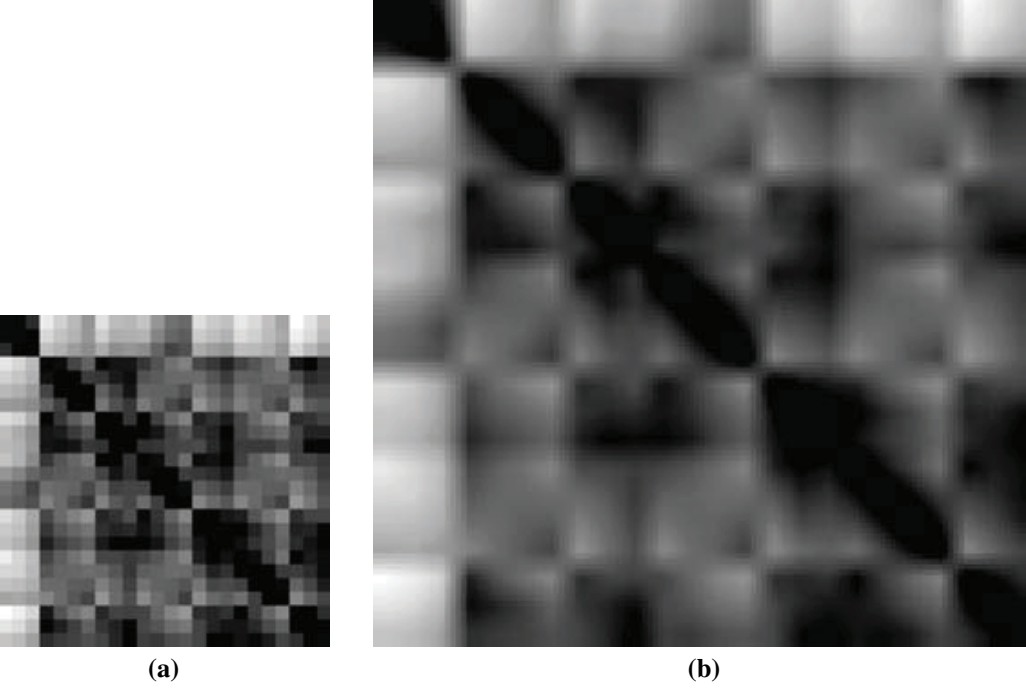

**Figure 5  Images of ligand structures: (A) non-scaled and (B) scaled.**

threshold values from the training data and uses these in test data prediction. We have used the same set of features that were generated and analyzed for the structural class prediction problem to solve the ligand-binding problem. In this section, we present the necessary materials and methods that were used for the ligand-binding problem.

### Ligand-binding dataset

We've used Computer Vision and Pattern Discovery for Bioimages Group @ BII as our dataset. In our dataset, 3,000 protein-ligand complexes were determined experimentally with 3D structures available. Each protein and its ligand are of one-to-one correspondence, that is, they can bind to each other and make Protein-Ligand complex. The dataset has 3,000 pairs of protein and ligand where the same name/ID of protein and ligand interacts/binds with each other.

We've used OpenCV (*Bradski & Kaehler, 2008*) library to create images from PDB files. For protein, we've considered the coordinates of only the alpha-carbons to generate the distance matrix to create an image. Because alpha-carbon can represent the structural information of protein quite well. But the given ligands were small in terms of atom number. So, while creating ligand images, we've considered all the atom's coordinates for generating distance matrix.

Among the PDB files, 33 ligands have only one atom, which will create a $1 \times 1$ image having no significance for feature extraction. So, we had to compromise those 33 ligands as well as 33 corresponding proteins from training. Figure 5 shows a sample non-scaled and scaled image of a ligand.

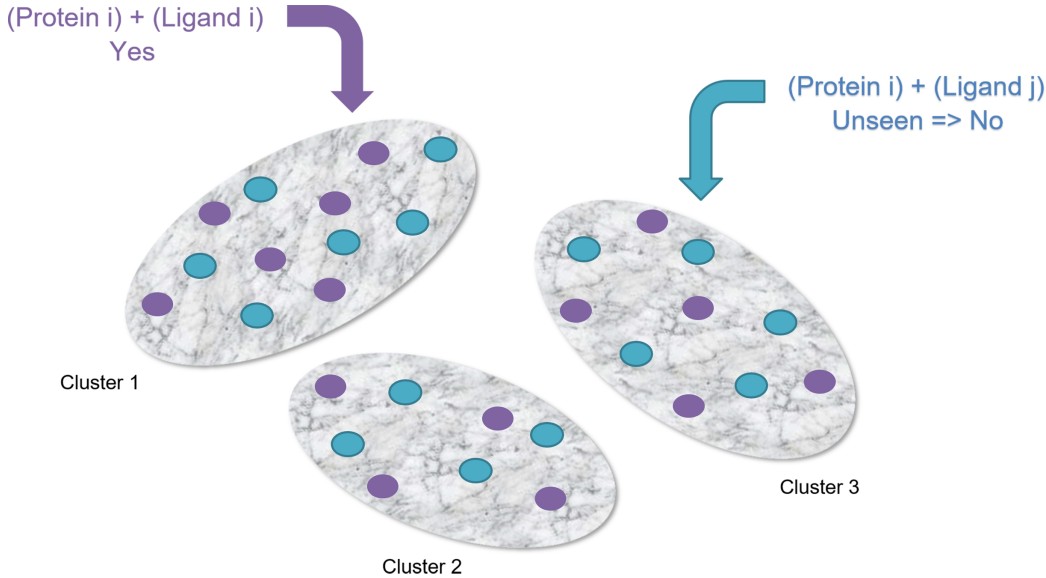

**Figure 6  Clustering-based undersampling.**

### Handling imbalance

The given dataset has only positive instances (the pairs of protein and ligand where they bind with each other). But there were no negative instances (the pairs of protein and ligand where they do not bind with each other). The missing negative instances have created our dataset highly imbalanced. To overcome this imbalance, we've generated negative instances in two different ways.

1. Random Negative Undersampling: We have 2,967 protein PDB and 2,967 ligand PDB where 8,803,089 pairs are possible. Among these, 2,967 pairs are given as positive instances and the rest 8,800,122 pairs are unknown/unseen instances. From the unseen pairs, we've taken 2,967 pairs randomly as negative instances to make our dataset balanced.

2. Clustering-Based Undersampling: Using the positive instances (2,967 pairs), we've created 10 clusters. Then we've searched for 2,967 unseen pairs randomly as negative instances where they belong to those 10 clusters. We've made sure that each cluster has the same number of positive and negative instances to make the dataset balanced (See Fig. 6).

### Similarity-based classifier

We've developed a similarity-based clustering method to predict the binding class. Distance is used to measure similarity. Our methodology is given in Fig. 7 and the pseudo-code in Algorithm 1.

From the PDB dataset of proteins and ligands, firstly we have generated images and converted to 128 × 128 images for each protein and ligand. From these images, we have generated two different features.

1. CoMOGrad and PHOG: CoMOGrad stands for Co-occurrence Matrix of the Oriented Gradient of Distance Matrices and PHOG stands for Pyramid Histogram of Oriented Gradient (*Karim et al., 2015*). This methodology also uses the α carbon distance matrix of protein. The dimension of all distance matrix is converted to 128 × 128. In CoMOGrad, the gradient angle and magnitude is computed from the distance matrix and the values are quantized. Quantization is a compressing technique which compresses a range of values to a single quantum value. In this methodology, the values are quantized to 16 bins which produce a co-occurrence matrix which is 16 × 16 matrix. The matrix is converted into a vector of size 256. Quadtree from the distance matrix is created with the desired level in PHOG. Gradient Oriented Histogram of each node is calculated with the preferred number of bins and bin size. In gradient oriented histogram an image is divided into small sub-images called cells and histogram of edge orientations are accumulated within the cell. The combined histogram entries are used as the feature vector describing the object. Total features which are the multiplication of total nodes and number of bins are incorporated in the vector with the size of the total number of features. The vector is normalized by dividing it with the sum of its components.

2. Hybrid Local Binary Pattern: LBP (*Ojala, Pietikainen & Harwood, 1994*) is a procedure of LBP histogram. We have used all the five feature groups described in the last section for structural class prediction problem.

Distance can only be calculated between proteins or between ligands. We've used KNN and Clustering method to calculate these distances.

1. RELATEDLIGANDS ($\mathbb{NP}$): For a given protein, find K-nearest proteins. The ligands those bind with the above nearest proteins are the Related Ligands for the given protein (See Fig. 8).

2. RELATEDPROTEINS ($\mathbb{NL}$): For a given ligand, find K-nearest ligands. The proteins those bind with the above nearest ligands are the Related Proteins for the given ligand (See Fig. 9).

To find the distances between pairs of ligands and proteins are calculated using Euclidean and Manhattan distances. Threshold is the boundary between similarity and dissimilarity in terms of distance. If distance is less than the threshold, then prediction is positive similarity, else the prediction is negative similarity. The threshold of each category of distances is the average distance based on the number of nearest neighbors. Measuring distance has been done in two ways. One method is to get the cluster mean of k-RelatedLigands/RelatedProteins, then measure the distance from the GivenProtein/GivenLigand. Another method is to measure distances between the GivenProtein/GivenLigand and k-RelatedLigands/RelatedProteins, then take the average of those distances as the final distance.

For a given pair of Protein and Ligand, we want to predict if the will bind with each other or not. For measuring distance $d_b$ from the given protein, we searched for k-nearest proteins and found the k related ligands accordingly. Then we've calculated the distance using the above-mentioned methods. Then we've taken the vote for the binding class by all

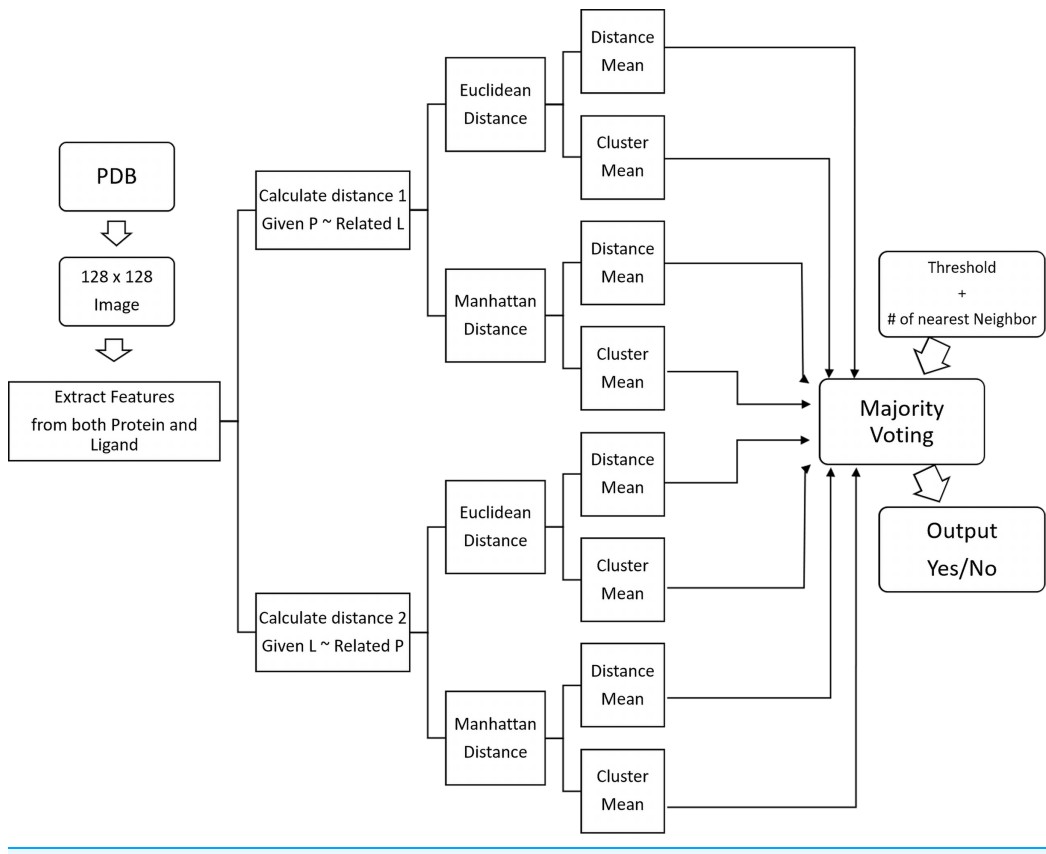

**Figure 7 Block diagram of similarity-based clustering.**

categories of distances based on their thresholds. Then finally, we've used majority voting mechanism to predict the binding class.

### Hyperparameters

There are some hyperparameters of our proposed method.

1. Number of nearest neighbors: Our algorithm's prediction accuracy is highly dependent on the number of nearest neighbors for finding both RELATEDLIGANDS ($\mathbb{NP}$) and RELATEDPROTEINS ($\mathbb{NL}$). We've used five and three (best fit) nearest neighbors in this experiment.

2. Threshold: This is the threshold of distance for determining whether two proteins or two ligands are similar or not. For a higher value of threshold, there is a higher possibility for our algorithm to predict positive binding class for the majority of the protein-ligand pairs. Also, the lower the threshold is, the higher is the possibility of negative binding class prediction. We have taken the average of distances among three nearest neighbors as our threshold for each category of the distances. In terms of setting the threshold values, distances of each training sample from the whole training data was measured. While doing so, negative pairs of protein-ligand were considered when positive pair of protein-ligand was given for getting distances. Similarly, positive pairs of protein-ligand were considered when negative pair of protein-ligand was given. The average of these distances was taken as threshold values.

**Algorithm 1** Similarity-based clustering algorithm.

**Data**: A pair $(p, l)$, a protein structure and ligand structure in PDB format

**Result**: Decision, whether they will interact or not

1 **for** *all proteins and ligands* **do**

2     generate images & extract features

3 **end**

4 **for** *each of the given pairs of protein-ligand* **do**

5     $\mathbb{NP} \leftarrow k\text{-NEARESTPROTEINS}(p)$ of the given protein

6     $\mathbb{RL} \leftarrow k\text{-RELATEDLIGANDS}(\mathbb{NP})$

7     $d_l \leftarrow$ distance between given ligand, $l$ & $\mathbb{RL}$

8     **if** $d_l < threshold_l$ **then**

9       $v_l \leftarrow$ vote for positive bind

10     **else**

11       $v_l \leftarrow$ vote for negative bind

12     **end**

13     $\mathbb{NL} \leftarrow k\text{-NEARESTLIGANDS}(l)$ of the given ligand

14     $\mathbb{RP} \leftarrow k\text{-RELATEDPROTEINS}(\mathbb{NL})$

15     $d_p \leftarrow$ distance between given protein, $p$ & $\mathbb{RP}$

16     **if** $d_p < threshold_p$ **then**

17       $v_p \leftarrow$ vote for positive bind

18     **else**

19       $v_p \leftarrow$ vote for negative bind

20     **end**

21     $v \leftarrow$ majority voting between $(v_l, v_p)$

22 **end**

23 **return** $v$

## RESULTS AND DISCUSSION

This section is the description of our experiments performed in this study. Some of the experiments were carried out in a personal desktop computer having Intel Core i3 and 4 GB RAM and others were experimented in a Computing Machine provided by CITS, United International University which was equipped with eight-core processors each having a Dell R 730 Intel Xeon Processor (E5-2630 V3) with 2.4 GHz speed and 18.5 GB memory. Java language was used for data preprocessing including feature generation using OpenCV software library, negative data generation and data merging using Eclipse IDE with Java 8 standard edition. Python language was used to implement our algorithm using the Spyder IDE. Weka tool was used to run the traditional classification algorithms for the comparison with our algorithm. 10-fold cross-validation method has been used to get the performance of our model.

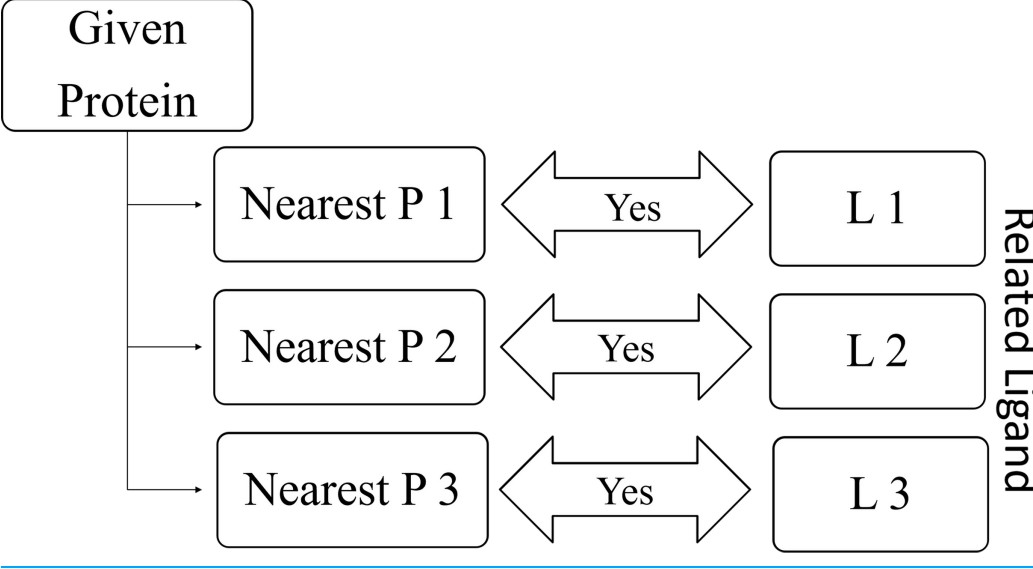

**Figure 8  Relation between given protein and related ligands.**

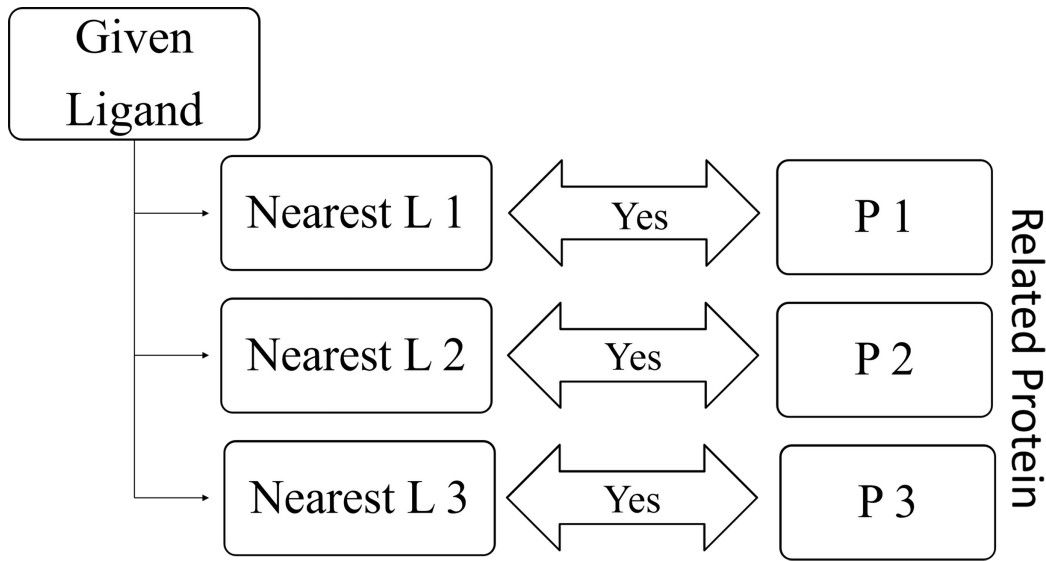

**Figure 9  Relation between given ligand and related proteins.**

We also used Scale-invariant feature transform (SIFT) (*Lowe, 2004*) methodologies in our experiments. Each descriptor has a 128-dimensional feature vector. The number of descriptors of SIFT from every image is not specific so we cannot use traditional machine learning techniques. Hence to apply traditional machine learning procedure and specify the feature vector, we have split the image into 16 slices and took one descriptor from each of the slice images. Therefore we got 2,048 number of attributes ($8 \times 16$) from each image. We have tested the dataset with the same classifiers mentioned in this paper.

**Table 3 Classifier accuracies for different types of features and groups of features.**

| Image type | Feature type | Classifiers | | | | |
|---|---|---|---|---|---|---|
| | | KNN | Naive Bayesian | SVM | Adaboost (J48) | Random forest |
| Scaled | A | 50.81 | 50.63 | 69.16 | 68.62 | 75.13 |
| Non scaled | A | 53.88 | 34.53 | 68.44 | 72.15 | 75.85 |
| Scaled | B | 21.97 | 8.58 | 17.45 | 24.86 | 30.83 |
| Non scaled | B | 30.83 | 25.40 | 23.23 | 37.25 | 42.85 |
| | C | 15.82 | 5.60 | 25.76 | 36.07 | 39.51 |
| Scaled | D | **69.43** | 48.01 | 67.63 | 70.97 | 74.77 |
| Scaled | E | 61.30 | **56.23** | 64.73 | 72.60 | 74.05 |
| Non scaled | AB | 51.71 | 32.45 | 68.89 | 71.33 | 73.50 |
| Non scaled | ABC | 27.30 | 32.27 | 68.62 | 69.80 | 74.23 |
| Non scaled + scaled | ABCD | 41.77 | 33.36 | 74.68 | 72.87 | 77.84 |
| Non scaled + scaled | ABCDE | 50.27 | 34.53 | **77.21** | **75.22** | **78.66** |

**Note:**
The highest accuracy among the feature types in each category of classifiers are marked in bold.

The results didn't turn up to be better or close to our proposed methodology in this literature. The result of accuracy on different classifiers using SIFT is given in File S1.

We've tried outlier detection method to handle the absence of negative data in Ligand-Binding dataset. But it gave overfitting problem. Details are in File S5.

## Analysis of features

A different sets of parameters were used for each classifier used in this research. A linear searching was used with no distance weighting for KNN. In the case of the Naive Bayesian Classifier, SVM, a polynomial kernel was used with $c = 1.0$ and $\varepsilon = 1.0 \ w^{-2}$. Data was normalized before supplying to the classifier. J48 decision tree classifier was used in Adaboost classifier as the weak base classifier. Classifier number of iterations was set to 100 for Random Forest.

We've used the "StratifiedRemoveFolds" filter which is available in Weka for creating Train and Test set. This is a supervised instance filtering process that takes a dataset and outputs a specified fold for cross-validation *Kohavi (1995)*. We've taken one random stratified fold out of 10 stratified folds as a Test set. The Train set has been generated by applying SMOTE to the rest of the nine folds for Class balancing. Then we've tested the performance of the features shown in Table 2 and different combinations of them using these Train & Test set. Accuracy scores of the feature groups are given in Table 3. Other performance metrics results (sensitivity, specificity, f1 score) can be found on File S7. No Cross-Fold Validation was performed here as the performance of one Test set is enough to find the best-performing group combination of features. The highest percentage of correctly classified instances achieved for each of the classifiers are indicated by the boldly faced values of the table.

After running the experiments for our five feature groups ABCDE classifies the highest percentage of correct instances in Random Forest, Adaboost(J48) and SVM among all

**Table 4 Comparison of the proposed features in this paper with *Karim et al. (2015)* for structural class prediction.**

| Performance metric | Features | Classifiers | | | | |
|---|---|---|---|---|---|---|
| | | KNN | Naive Bayesian | SVM | Adaboost (J48) | Random forest |
| Accuracy | *Karim et al. (2015)* | **68.03** | **56.40** | 76.52 | 72.50 | 74.27 |
| | This paper | 51.37 | 35.35 | **77.27** | **76.25** | **76.76** |
| Sensitivity | Karim | **68.04** | **56.41** | 76.53 | 72.5 | 74.28 |
| | This paper | 51.37 | 35.34 | **77.28** | **76.25** | **76.76** |
| Specificity | *Karim et al. (2015)* | **92.98** | **89.49** | 94.64 | 92.78 | **93.41** |
| | This paper | 92.93 | 84.09 | **95.04** | **93.02** | 93.34 |
| Score | *Karim et al. (2015)* | **69.15** | **56.63** | 77.28 | 72.67 | 74.36 |
| | This paper | 54.21 | 31.88 | **78.28** | **76.23** | **76.77** |

**Note:**
The highest score of each performance metric between the features among each category of classifiers are marked in bold.

other feature groups. Feature scaled E and D individually provides the highest accuracy in Naive Bayesian and KNN. As the whole combination of all feature groups, accuracy gives the highest percentage than any other feature group, thus we conclude that the best-performing feature group combination is ABCDE and the best classifier is SVM classifier.

## Effectiveness in structural class prediction

In this section, we compare the performance of our proposed method with CoMOGrad and PHOG (*Karim et al., 2015*). For comparison with our methodology in this literature, we applied CoMOGrad and Phog techniques and Wavelet and Pyramid Histogram techniques in our dataset of 11,052 instances. We conducted experiments with different classifiers using the same parameters as we did for feature analysis with the feature groups. We've discussed in the Analysis of Features section that we've tested our feature groups using StratifiedRemoveFolds and finally selected ABCDE as our best feature group and no Cross-Fold Validation was done. After selecting the feature group, we have had to apply Cross-Fold Validation to use individual observation exactly once for validation. For performing 10-Fold Cross-Validation, we've generated 10 different folds using "StratifiedRemoveFolds" and in 10 iterations, one fold has been kept as Test set and SMOTE has been applied to the other nine folds to get the Train set of the running iteration. The result of performance metric (accuracy, sensitivity, specificity and f1 score) for each stratified iteration is given in File S2 for HybridLBP and File S3 for ComogPHOG. SMOTE hasn't been applied before splitting the data for cross-validation to avoid the presence of artificial data in the test set. Average scores of the 10 iterations are the actual score for comparing performance. This cross-validation was applied to all benchmark dataset along with ours to establish a valid comparison. The results are given in Table 4. From Table 4 it can be comprehended that our feature group ABCDE outperforms CoMOGrad and PHOG in Random Forest, Adaboost & SVM. CoMOGrad and PHOG surpassed our feature groups in KNN and Naive Bayesian. It can be noted that the combination of our feature groups are three-fourths of CoMOGrad and PHOG.

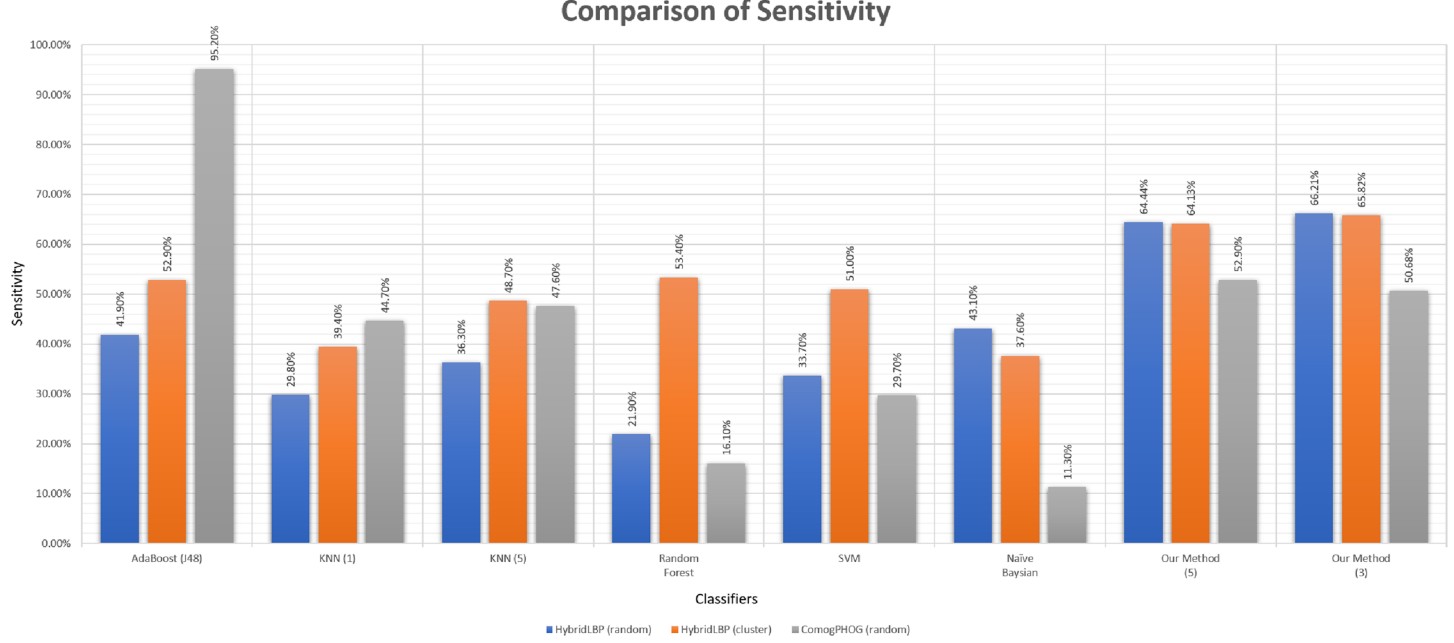

**Figure 10** Barplot showing the performance of different algorithms on ligand-binding dataset.

**Table 5 Sensitivity, specificity and F1 score comparison among different methods for ligand-binding prediction.**

| Performance metric | Features | Adaboost (J48) (%) | KNN (5) (%) | Random forest (%) | SVM (%) | Naive Bayesian (%) | Our method (3) (%) |
|---|---|---|---|---|---|---|---|
| Sensitivity | HybridLBP (random) | 41.90 | 36.30 | 21.90 | 33.70 | 43.10 | **66.21** |
| | HybridLBP (cluster) | 52.90 | 48.70 | 53.40 | 51.00 | 37.60 | **65.82** |
| | ComogPHOG (random) | **95.20** | 47.60 | 16.10 | 29.70 | 11.30 | 50.68 |
| Specificity | HybridLBP (random) | 41.20 | 47.20 | 18.20 | 39.00 | **49.60** | 42.19 |
| | HybridLBP (cluster) | 55.80 | 58.00 | 62.20 | 56.90 | **65.50** | 42.19 |
| | ComogPHOG (random) | 3.10 | 47.50 | 13.00 | 38.80 | **83.90** | 59.85 |
| F1 score | HybridLBP (random) | 41.80 | 38.40 | 21.50 | 34.60 | 44.60 | **59.06** |
| | HybridLBP (cluster) | 53.70 | 51.10 | 55.80 | 52.50 | 43.70 | **58.83** |
| | ComogPHOG (random) | **65.20** | 47.60 | 15.80 | 31.10 | 17.80 | 53.11 |

**Note:**
The highest score of each performance metric between the classifiers among each category of features are marked in bold.

It also can be discerned that the accuracy percentage in SVM is higher than all the classifier results. Thus, our novel features are small in size and can classify instances more accurately than CoMOGrad and PHOG.

We have revealed the precedence of our methodology over CoMOGrad and PHOG (*Karim et al., 2015*).

## Effectiveness in ligand-binding prediction

Sensitivity is the true positive rate regarding the positive instances. As we had to generate the negative data artificially, sensitivity is the actual scale of performance measuring as

the positive data were the actual data. Using the thresholds gained using the negative data, the sensitivity of our algorithm is very good comparing to other existing algorithms shown in Table 5 and Fig. 10.

We have generated three different datasets based on three different features. Hybrid LBP gives 736 long feature vectors from protein images and 677 long feature vectors from ligand images. So, for one protein-ligand pair, we've got 1,413 (736 + 677) attributes and one Binding Class value as one instance. The above mentioned two types of negative data (random and Clustering-Based Undersampling) were generated using Hybrid LBP for balancing the data. CoMOGrad and PHOG gives 1,021 or 1,020 long feature vectors from protein image, but for ligand images, it gives 1,020 long feature vectors. We assumed "0" as the last feature in protein where features were 1,020 long, to make it a 1,021 long feature. So, for one protein-ligand pair we've got 2,041 (1,021 + 1,020) attributes and one Binding Class value as one instance. Random negative undersampling was used in CoMOGrad and PHOG but Clustering-Based Undersampling was not possible as some clusters couldn't get any unseen pairs of protein and ligand. Our method was used based on three nearest neighbors and shown on the above table and chart.

Accuracy is not shown because it is dependent on the artificial negative data. However, specificity is shown to correlate with sensitivity, not to judge the algorithms as it is a true negative rate. On the other hand, the F1 score is also valuable because of being a harmonic mean of sensitivity & precision where both scores are based on positive data. For more details, please check the File S6.

We can see that AdaBoost works better than our algorithm in terms of sensitivity in ComoGrad and PHOG dataset. Because Ligand data were so small in terms of the number of atoms that ComoGrad and PHOG gave zeros for most of the ligands. This is an overfitting problem as high sensitivity is offset by low specificity. But our algorithm's overall performance is better than other machine learning algorithms in the three different feature datasets.

## CONCLUSIONS

In this paper, we showed how accurately we can detect protein classes using the combination of different image-based feature groups generated from protein images. We also propose a simple similarity-based clustering method to predict protein-ligand binding without using deep-learning or neural-networks. This simple distance-based algorithm is quite effective compared to complex machine learning algorithms. Our main limitation was the missing negative data. If we had the actual negative data, we could've determined the perfect thresholds for each category of distances, and that would give us a more accurate prediction. Another problem was the dimensions of small Ligands as we're using image-based features. As the advancement of deep learning, neural network, and many other deep learning techniques are being used to classify images, many remarkably interesting applications can be made. For our future advancement, we wish to introduce new features to improve accuracy, use new tools and explore other fields of computer vision such as human emotion detection. Besides, we will try to extract some unique

features from the ligand dataset so that the dimensionality problem does not affect our protein-ligand binding prediction.

## ACKNOWLEDGEMENTS

We thank Rezaul Karim for sharing the SQL dataset files and algorithms to generate Distance Matrix from PDB files for CoMOGrad and PHOG.

### Funding

The authors received no funding for this work.

### Competing Interests

The authors declare that they have no competing interests.

### Author Contributions

- Nafees Sadique conceived and designed the experiments, performed the experiments, analyzed the data, performed the computation work, prepared figures and/or tables, authored or reviewed drafts of the paper, and approved the final draft.
- Al Amin Neaz Ahmed conceived and designed the experiments, performed the experiments, analyzed the data, performed the computation work, prepared figures and/or tables, authored or reviewed drafts of the paper, and approved the final draft.
- Md Tajul Islam performed the computation work, prepared figures and/or tables, authored or reviewed drafts of the paper, and approved the final draft.
- Md. Nawshad Pervage performed the computation work, prepared figures and/or tables, authored or reviewed drafts of the paper, and approved the final draft.
- Swakkhar Shatabda conceived and designed the experiments, analyzed the data, prepared figures and/or tables, authored or reviewed drafts of the paper, and approved the final draft.

### Data Availability

Data and code are available at: https://github.com/NafeesSadique/Image-based-effective-feature-generation-for-Protein-Structural-Class-and-Ligand-Binding-prediction.

### Supplemental Information

Supplemental information for this article can be found online at http://dx.doi.org/10.7717/peerj-cs.253#supplemental-information.

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
