# Peer review of "Image-based effective feature generation for protein structural class and ligand binding prediction"

_PeerJ Computer Science, doi:10.7717/peerj-cs.253_

## Round 0.1 · original submission · Major Revisions

I have received the review reports for your paper submitted to PeerJ Computer Science from the reviewers. According to the reports, I will recommend major revision to your paper. Please refer to the reviewers’ opinions to improve your paper (sspecially you need to have a good reply to referee 2). Please also write a revision note such that the reviewers can easily check whether their comments are fully addressed. We look forward to receiving your revised manuscript soon.

Reviewer 1 ·

Basic reporting

The topic of the paper is interesting and the problems are significant, although I am not too familiar with the relevant bioinformatics field. One of the key references (Karim et al) use a similar technique for retrieving similar proteins as opposed to classification, so it would be good if the authors can clarify their motivation and whether classification is a significant topic. Following the paper was difficult in places because the sequence of topics is not always logical, e.g. SIFT features are suddenly mentioned at the end of the paper after not being mentioned at all before. The paper does need a more logical organisation.

Overall I found the quality of the paper is at a draft stage and needs more work before publication. There are many, many improvements still to be made. Some of them, but not all:
-- novel contributions should be made clear, and distinguished from already existing contributions (currently they are all over the place)
-- Referencing needs to be fixed, e.g. Mohri et al did not invent all of those ML algorithms, please put the proper sources down
-- for the ligand binding problem no example images are given so it's not clear how they look compared to the others
-- many things need to be explained better, e.g. how does the Karim et al method actually generate images? how are points arranged along the x/y axis and how does this affect the accuracy? what was the distribution of original image scales before scaling and is scaling significant?
-- Gabor filter is not fully explained (at least, only one of the wikipedia formulas are given in the paper)
-- Fig 4 seems to be wrong, i.e. the numbers going from (b) to (c) are divided by 10, not 100 which the text states
-- motivation for the new LBP features should be given. Why did the authors think these modifications are good?

Experimental design

In terms of experiments, I am not convinced that the right methodology was followed. SMOTE was used to artificially balance the dataset for the 5 class problem. However this appears to have been done *before* splitting the data for cross validation. Therefore the training data contains artificial examples derived the test data which is not fair. For the ligand binding dataset, one class or outlier detection methods could be used since there is no true negative training data. And the high sensitivity in Table 5 is probably offset by low specificities. The authors should provide both results as well as F1.

Validity of the findings

See answer above. Experimental design impacts validity.

Additional comments

Overall the paper is interesting but I think in it's current form it is not clearly described enough to be reproduced, the evaluation needs improvement, and the paper needs more logical organisation and improvements in presentation.

Reviewer 2 ·

Basic reporting

See general comments

Experimental design

See general comments

Validity of the findings

See general comments

Additional comments

Unfortunately, this study makes no sense. Protein structures are converted into distance matrices, which are used to “predict” the structural class. You know the structural class directly from the structure, so why converting into images and then predicting the class? Furthermore, what is the point of assigning known structures to 7 major SCOP classes (all alpha, all beta, etc)? SCOP, CATH, ECOD and other classification systems are much more sophisticated and include evolutionary relationships. These resources are widely to study various aspects of protein structure and function. The proposed method does not offer any unique capabilities and I just cannot see how it could be useful to anyone.

Reviewer 3 ·

Basic reporting

The authors had taken an effort to classify and predict protein structures and ligand binding.

The article should include sufficient introduction and relevant prior literature should be appropriately referenced.

All appropriate raw data have been made available.

Experimental design

Validation of the algorithm is not clear. Sensitivity and specificity values and correlation for ML should have been furnished.

Validity of the findings

No comment

Additional comments

The introduction part needs to be improved. Ligand definitions are not clear.

Comparison with the existing methods thereby the novelty be established.

Validation of the algorithm is not clear. Sensitivity and specificity values and correlation for ML should have been furnished.

Important Features identified and their influence to the prediction and the significance needs to be discussed.

Case studies with a protein for class and ligand binding may be given.

---

## Round 0.2 · accepted · Accept

According to the review comments, I will accept the paper for publication.

Reviewer 3 ·

Basic reporting

No comment

Experimental design

No comment

Validity of the findings

No comment

Additional comments

Tittle: Image-based effective feature generation for Protein Structural Class and Ligand Binding prediction
Authors: Nafees Sadique et al.

In general the authors have made a tremendous effort to improve the article.
Data analysis and prediction are included.
However, if the tool developed is live then it will be of great help to the users.
Good luck!